REGISTERED REPORT

# Registered report: the microRNA miR-34a inhibits prostate cancer stem cells and metastasis by directly repressing CD44

Jia Li[1], Matthew Lam[2], Reproducibility Project: Cancer Biology*

[1]Crown Bioscience, Santa Clara, California, United States; [2]Breakthrough Breast Cancer, London, United Kingdom

*For correspondence:
fraser@scienceexchange.com

Group author details
Reproducibility Project: Cancer Biology
See page 21

Competing interests:
See page 21

**Abstract** The Reproducibility Project: Cancer Biology seeks to address growing concerns about reproducibility in scientific research by conducting replications of selected experiments from a number of high-profile papers in the field of cancer biology. The papers, which were published between 2010 and 2012, were selected on the basis of citations and Altimetric scores (*Errington et al., 2014*). This Registered report describes the proposed replication plan of key experiments from 'The microRNA miR-34a inhibits prostate cancer stem cells and metastasis by directly repressing CD44' by Liu and colleagues published in *Nature Medicine* in 2011 (*Liu et al., 2011*). Liu and colleagues first demonstrated that miR-34a levels were reduced in CD44+ prostate cancer cells (Figure 1B). They then showed that xenograft tumors from cells expressing exogenous miR-34a were smaller in size than control tumors (Supplemental Figure 5C). Tumors with exogenous miR-34a showed reduced levels of CD44 expression (Figure 4A), and mutation of two putative miR-34a binding sites in the CD33 3′ UTR partially abrogated signal repression in a luciferase assay (Figure 4D). The Reproducibility Project: Cancer Biology is a collaboration between the Center for Open Science and Science Exchange, and the results of the replications will be published by *eLife*.

## Introduction

Many cancers display characteristic growth patterns associated with cancer stem cells (CSCs). In recent years, the importance of microRNAs in the initiation and maintenance of various cancers has become more widely explored (*Maroof et al., 2014*). In particular, miR-34 family members (miR-34a, miR-34b and miR-34c) have been identified as playing a role in the p53 pathway as well as influencing Notch and Met signaling (*Misso et al., 2014*), and are associated with inhibition of glioblastoma (*Abel et al., 2009*; *Wiggins et al., 2010*), pancreatic tumor cells (*Corney et al., 2010*) and gastric cancer (*Ji et al., 2008*). In their 2011 *Nature Medicine* paper, Liu and colleagues elucidated the importance of a particular microRNA, miR-34a, on the action of CD44+ prostate cancer putative CSCs, demonstrating that miR-34a was barely expressed in CD44+ CSCs and had tumor suppressing properties. They also showed that miR-34a affected the levels of CD44, a widespread marker of putative cancer stem cells (*Liu et al., 2011*).

In Figure 1B, Liu and colleagues purified CSCs from three xenograft cancer models; LAPC9, LAPC4 and Du145 cells. They assessed the levels of miR-34a expression as a percentage of expression from CD44+ cells vs CD44− in each model by quantitative RT-PCR, as compared to levels of let-7b, a known tumor suppressive miRNA that is underexpressed in CD44+ cells. They found that miR-34a levels were markedly low in these xenograft models as compared to let-7b levels. This experiment will be replicated in Protocol 3.

In Supplemental Figure 5C, the authors demonstrated that the addition of exogenous miR-34a—through lentiviral infection of LAPC4 cells—decreased the size of resultant xenograft tumors, as compared to LAPC4 cells infected with a control lentivirus. This experiment demonstrates the functional relevance of miR-34a, and will be replicated in Protocol 4.

In Figure 4, Liu and colleagues present data supporting the hypothesis that CD44 is a target of miR-34a. In Figure 4D, they demonstrate that mutation of two putative miR-34a binding sites in the 3′ UTR of CD44 decreased signal in an in vitro luciferase assay (replicated in Protocol 7). They show evidence that tumors overexpressing miR-34a by lentiviral infection with miR-34a (Supplemental Figure 4A, replicated in Protocol 6) have reduced expression of CD44 (Figure 4A, right panel (Western blots), conceptually replicated in Protocol 5).

Since the publication of Liu and colleagues' work, additional groups have identified roles for miR-34 family members in gallbladder cancer (*Jin et al., 2013*), mesothelioma (*Toyooka, 2013*) and breast cancer (*Achari et al., 2014*). In particular, Basak and colleagues explored the relationship of CD44 and miR-34a in breast cancer cells and showed that addition of exogenous miR-34a to CD44$^{hi}$ cells reduced their colony forming efficiency (*Basak et al., 2013*). Fan and colleagues showed that miR-34a reduced the metastatic potential, invasion and migration of osteosarcoma cells (*Fan, 2013*). Siemens and colleagues also provide evidence that miR-34a regulates the expression of CD44 (*Siemens et al., 2013*, *2014*).

## Materials and methods

Unless otherwise noted, all protocol information was derived from the original paper, references from the original paper, or information obtained directly from the authors. An asterisk (*) indicates data or information provided by the Reproducibility Project: Cancer Biology core team. A hashtag (#) indicates information provided by the replicating lab.

### Protocol 1: Maintenance of LAPC4 xenograft prostate tumors

This protocol describes the maintenance of the LAPC4 xenograft tumor tissue by serial implantation in NOD/SCID mice.

### Sampling

- No power calculations required.

### Materials and reagents

| Reagent | Type | Manufacturer | Catalog # | Comments |
|---------|------|--------------|-----------|----------|
| LAPC4 cells | Cells | Provided by original authors | | |
| 2 mm × 4 mm die set | Instrument | International Crystal Laboratories | – | – |
| NOD/SCID mice, male, 6–8 weeks old | Mice | Jackson Laboratories | 001303 | – |
| Ketamine HCl | Reagent | Jiang Su Heng Rui Medicine Co. | H32022820 | Replaces Cat. No. 10177 from Henry Schein Animal Health |
| Xylazine | Reagent | Huamu Animal Health Products Co. | (2009) 070011582 | Replaces Cat. No. 10177 from Henry Schein Animal Health |
| IsoFlo isoflurane | Reagent | Henry Schein Animal Health (Abbott) | 017579 | Optional |
| Betadine solution | Reagent | Henry Schein Animal Health (Purdue Frederick) | 038248 | – |
| Testosterone propionate powder | Reagent | Sigma–Aldrich | T-1875 | – |
| Flunixiject 50 mg/ml (banamine) | Reagent | Henry Schein Animal Health (Bimeda) | 14165 | – |
| Ethanol | Reagent | Specific brand information will be left to the discretion of the replicating lab and recorded later | | |
| Sterile saline | Reagents | Specific brand information will be left to the discretion of the replicating lab and recorded later | | |
| Matrigel matrix, high concentration | Reagent | BD | 354248 | – |
| IMDM with GlutaMAX, HEPES and sodium pyruvate | Medium | Gibco/Life Technologies | 31980-030 | This reagent was referred to as 'DMEM' in the original study |
| FBS | Reagent | Sigma–Aldrich | F2442 | Replaces Benchmark Cat. #100-106 |
| Insulin needle and syringe | Materials | BD | 320310 | – |

## Procedure

Notes:

- This protocol contains information described in *Li et al. (2009)*.
- *Based on the number of tumor cells provided by the original authors, inject at least two mice in order to maintain the tumor tissue and provide sufficient cells for downstream experiments.

1. Anesthetize 6–8 week old male NOD/SCID mice with Ketamine/xylazine.
   a. #Ketamine (50 mg/ml) and xylazine (20 mg/ml) are mixed with saline (2:0.42:5.91), sterilized with 0.22 µm filter and 0.08–0.1 ml is injected per mouse.
   b. The anesthesia may be supplemented with inhalant isoflurane mixed 50/50 with mineral oil when the mouse responds to squeezing of the hind foot or shows response during the procedure.
2. Shave the dorsal hair of each mouse and prep the skin with Betadine and then 70% ethanol. Make a small 0.5 cm incision in the dorsal skin of the mouse, right under the head. Open a pocket under the skin with surgical scissors and insert a prepared testosterone pellet. Staple the incision to close.
   a. Prepare the testosterone pellet by directly compressing testosterone propionate powder with a 2 mm × 4 mm die set.
3. Immediately after testosterone pellet implantation, inject tumor cells into mice.
   a. Resuspend $2 \times 10^6$ cells in 30 µl IMDM supplemented with 10% FBS and chill on ice.
   b. Mix with 30 µl concentrated Matrigel.
   c. Inject the mixture (60 µl total) into the flank of the mouse using an insulin syringe.
   d. Implant at least 2 and if possible 3 mice with tumor cells.
4. Inject the mouse subcutaneously with 0.02 ml of banamine followed by 1 ml of sterile saline to aid in recovery from the anesthesia. Put the mouse cage on a heating pad. Monitor the mouse until it starts to move.
5. After 7 days of surgery, remove the surgical staples.
6. Monitor tumor development starting from the second week after the surgery.
   a. Measure tumor incidence (number of tumors/number of injections).
   b. Measure latency (time from injection to detection of palpable tumors).
   c. Measure tumors weekly with caliper measurements (height, width, and depth to determine volume by the formula $V = 1/2*[\text{length} \times \text{width}^2]$).
7. Terminate animals at 4–6 months after tumor cell injection when tumor size reaches 1.5 cm in diameter and aseptically dissect out tumors.
   a. Lab will record euthanasia method.
   b. Weigh tumors.
8. Mince each dissected tumor into 1–2 mm$^3$ pieces using a razor blade.
   a. Use the tumor tissue for purifying LAPC4 cells (Protocol 2).

## Deliverables

- Data to be collected:
  ○ Record LAPC4 xenograft tumor maintenance (passages, tumor incidence, latency, tumor caliper measurements (and calculated volume), tumor weight at harvest, time of harvest, and euthanasia method, calculated density of each tumor based on volume and weight).
- Sample delivered for further analysis:
  ○ Minced LAPC4 tumor tissue for purification of LAPC4 cells. (Protocol 2).

## Confirmatory analysis plan

- Statistical Analysis of the Replication Data:
  ○ None required.
- Meta-analysis of original and replication attempt effect sizes:
  ○ None required.

## Known differences from the original study

The replicating lab will use a thoracentesis puncture needle to implant tumor tissue into recipient mice instead of making incisions. The replicating lab routinely uses this technique to implant tumor tissue into mice.

## Provisions for quality control

Tumor development characteristics will be recorded for each mouse. All data obtained from the experiment—raw data, data analysis, control data and quality control data—will be made publicly available, either in the published manuscript or as an open access dataset available on the Open Science Framework (https://osf.io/gb7sr/).

# Protocol 2: Purification of LAPC4 cells from xenograft tumors

This protocol describes how to isolate and separate human LAPC4 cells from xenografted tumors for further use in downstream protocols.

## Sampling

- No power calculations required.

## Materials and reagents

| Reagent | Type | Manufacturer | Catalog # | Comments |
|---|---|---|---|---|
| MACS separator and MACS MS column | Equipment | Miltenyi Biotec | 130-042-201 | – |
| MACS lineage cell depletion kit | Kit | Miltenyi Biotec | 130-090-858 | – |
| 40 µm cell strainer | Material | BD Falcon | 352340 | Original brand not specified |
| IMDM with GlutaMAX, HEPES and sodium pyruvate | Medium | Gibco/Life Technologies | 31980-030 | This reagent was referred to as 'DMEM' in the original study |
| IMDM with HEPES and sodium pyruvate | Medium | Gibco/Life Technologies | 12440053 | – |
| FBS | Reagent | Sigma–Aldrich | F2442 | Replaces Benchmark Cat. #100-106 |
| Accumax cell dissociation solution | Reagent | Innovative Cell Technologies | AM 105 | – |
| Erythrosine B | Reagent | ATCC | 30-2404 | – |
| Histopaque-1077 | Reagent | Sigma–Aldrich | 10771 | – |
| Insulin | Reagent | Sigma–Aldrich | I-6634 | – |
| Bovine serum albumin (BSA) | Reagent | Sigma–Aldrich | 05470 | Original brand not specified |
| Dulbecco's phosphate buffered saline (PBS); without $Ca^{2+}$ and $Mg^{2+}$ | Reagent | Sigma–Aldrich | D8537 | Original brand not specified |
| PE labeled anti-human EpCAM | Antibody | BioLegend | 324206 | – |
| Isotype control antibody | Antibody | BioLegend | 400314 | – |
| BD stain buffer | Reagent | BD Pharmingen | 554656 | – |
| FACSCalibur | Equipment | BD Biosciences | – | Original unspecified |

## Procedure

Note: This protocol contains information described in papers from *Patrawala and colleagues (2005)*, (*2006*), (*2007*); *Li and colleagues (2009)*.

1. Use tumor aseptically dissected out of animals in Protocol 1. Weigh tumor tissue in a pre-tared sterile Petri dish and record weight. Mince tumor into ~1 mm³ pieces using a razor blade in 1–2 ml cold IMDM + 15% FBS.
2. Transfer cell slurry to a 50 ml tube using a cut 1 ml pipette tip and add ~20× the volume of cold IMDM with GLUTAMAX + 10% FBS. Invert tube to mix and centrifuge at 130×*g* for 5 min at 4°C. Discard supernatant.
3. Rinse twice in cold IMDM with GLUTAMAX + 10% FBS and once in PBS to wash out serum. Invert tube to mix and centrifuge at 130×*g* for 5 min at 4°C. Discard supernatant.
4. Incubate slurry in 1× Accumax Cell Dissociation solution at 10 ml per 0.5 g tumor tissue in PBS for 30 min at room temperature under rotating conditions.
   a. Before experiment, bring Accumax to room temperature.

5. Briefly vortex tissue solution and allow residual undissociated tumor pieces to precipitate to the bottom of tubes for ~2 min. Collect cells from the supernatant into new tube.
   a. Residual tumor pieces can be subjected to one or more rounds of Accumax digestion (step 3). Store already-digested cells on ice. *Repeat until no large undissociated pieces remain.
   b. Pool dissociated cells.
6. Spin pooled cells at 130×g for 5 min.
7. Filter supernatant through a 40 µm cell strainer to obtain a cell suspension.
   a. Pre-wet strainer with medium.
   b. Collect into a 50 ml centrifuge tube and resuspend in 1–3 ml IMDM with GLUTAMAX + 10% FBS.
8. Count viability of cells on a hemocytometer using erythrosine B.
   a. If a large number of cells are obtained (>50 million live cells), then only a portion of that number should be used for further processing (flow cytometry time and sorting speed will put a limit on analyzing a maximum of ~20 million cells).
   b. Adjust cell suspension to 1–1.5 million cells (dead + live) per ml with IMDM with GLUTAMAX + 10% FBS.
9. Load 3 ml of cell suspension gently onto a 3 ml layer of Histopaque-1077 gradient in a 15 ml tube and centrifuge at 400×g for 30 min at room temperature.
   a. Load with a 1 ml pipette tip.
   b. For larger volumes, 15 ml of cell suspension can be added to 15 ml of Histopaque in a 50 ml tube at a concentration of no more than $1 \times 10^6$ cells/ml (dead + live).
10. Collect live nucleated epithelial cells at the interface of the two layers (opaque layer) using a 1 ml pipette tip to transfer to a new tube. Spin at 130×g for 5 min at RT.
    a. If multiple tubes were used in Step 9, combine the live cells from all tubes before spinning.
11. Resuspend cells in 0.1–5 ml of IMDM with GLUTAMAX + 10% FBS, count the cell number, and spin again at 130×g for 5 min at RT.
    a. Wash cells again as per Miltenyi's instructions.
12. Deplete cell mixture of lineage-positive host (mouse) cells using the MACS Lineage Cell Depletion Kit. Refer to additional details in protocol from Miltenyi. (Volumes given are for 5 million cells.)
    a. Suspend cell pellet in 40 µl of staining buffer (PBS, pH 7.2, 0.5% FBS, 5 µg/ml insulin, 0.5% BSA).
    b. Add 10 µl of supplied biotin-antibody cocktail (CD5, CD45R, CD11b, anti-Ly-6G, 7-4, and Ter-119).
    c. Mix well and incubate for 10 min at 4°C in the dark.
    d. Add 30 µl of staining buffer.
    e. Add 20 µl supplied anti-biotin microbeads.
    f. Mix well and incubate for additional 15 min at 4°C in the dark.
    g. Wash cells by adding 10–20× labeling volume in cold PBS and centrifuge at 130×g for 5 min at 4°C. Pipette off supernatant completely.
    h. Resuspend up to $10^7$ cells in 500 µl staining buffer.
    i. Assemble MACS apparatus by attaching the MACS Separator on the stand and placing the MACS MS column in the magnetic field with a 15 ml collection tube below the column.
    j. Prepare the column by rinsing with 500 µl of staining buffer.
    k. Apply cell suspension to the column. Allow cells to pass by gravity and collect the effluent as fraction with unlabeled cells, representing the enriched human tumor cell fraction depleted of mouse cells.
    l. Wash the column with 500 µl of staining buffer and collect the effluent in the same tube as effluent of step k. Perform 3×, each time once the column reservoir is empty.
    m. Centrifuge the effluent at 130×g for 5 min at 4°C and discard supernatant.
13. #Confirm the purity of the cells using FACS.
    a. Prepare the PE-labeled human EpCAM following the manufacture's instruction for the primary and isotype staining, calculating enough total volume to add 100 µl per sample Ab stain in cold Stain buffer.
    b. Use cold Stain buffer (PBS, pH 7.2, 0.5% FBS, 0.5% BSA) when preparing primary antibodies and isotypes.
    c. Add 100 µl per sample of diluted antibodies to the 5 ml FACS tube and resuspend cells by gently pipetting.

 d. Incubate for 45 min at 4°C, protected from light.
 e. Add 200 µl cold Stain Buffer to each tube and centrifuge at 1300 rpm (240×$g$) × 5 min at 4°C.
 f. Remove supernatant and resuspend in cold BD Stain Buffer at 200 µl/well and then pellet at 1300 rpm (240×$g$) × 5 min at 4°C. Repeat wash two times.
 g. Add 100 µl cold Stain Buffer/tube to resuspend cells and analyze samples immediately using BD FACSCalibur. Cell debris and cell aggregates will be gated out based on the FSC/SSC parameters.
14. Resuspend cells in 0.1–5 ml of IMDM with GLUTAMAX + 10% FBS and count the cell number.
15. Use resuspended cells for further analysis (Protocol 3, Protocol 4, and Protocol 7).

## Deliverables

- Data to be collected:
  ○ Weight of tumor tissue.
  ○ Cell viability counts before and after Histopaque gradient and after MACS depletion.
  ○ All FACS plots from assessing the purity of the cells.
  ○ Percentage of LAPC4 cells in enriched cell population.
- Sample delivered for further analysis:
  ○ Resuspended purified LAPC4 cells (for Protocols 3, 4 and 8).

## Confirmatory analysis plan

- Statistical Analysis of the Replication Data:
  ○ None required.
- Meta-analysis of original and replication attempt effect sizes:
  ○ None required.

## Known differences from the original study

- None noted.

## Provisions for quality control

The purity of the cells after depletion of lineage-positive cells will be determined by FACS. All data obtained from the experiment—raw data, data analysis, control data and quality control data—will be made publicly available, either in the published manuscript or as an open access dataset available on the Open Science Framework (https://osf.io/gb7sr/).

## Protocol 3: Expression of miR-34a in LAPC4 sub-populations

This protocol describes how to assess expression levels of miR-34a by qRT-PCR in isolated LAPC4 xenograft tumor cells, as seen in Figure 1B.

## Sampling

- The original data presented is qualitative. Based on power calculations with a range of possible variance, we will perform the experiment three times for a final estimated power of at least 82.2%.
  ○ See Power calculations for details.
- Each experiment consists of two cohorts:
  ○ Cohort 1: CD44$^+$ LAPC4 cells.
  ○ Cohort 2: CD44$^-$ LAPC4 cells.
  ○ Within each cohort, assess levels of three miRNAs:
    ■ Let-7b.
    ■ miR-34a.
    ■ miR-103.
      • These qRT-PCR reactions are run in technical triplicate.

## Materials and reagents

| Reagent | Type | Manufacturer | Catalog # | Comments |
|---|---|---|---|---|
| FITC-conjugated monoclonal anti-CD44 (clone G44-26) | Antibody | BD Pharmingen | 555478 | – |
| FITC-mouse IgG2b isotype control immunoglobulin | Antibody | BD Pharmingen | 555742 | – |
| MX3005P Real-time PCR system | Equipment | Stratagene | – | Replaces ABI Prism 7900 SDS |
| IMDM with GlutaMAX, HEPES and sodium pyruvate | Medium | Gibco/Life Technologies | 31980-030 | This reagent was referred to as 'DMEM' in the original study |
| miR-34a miRNA assay kit | Primer | Applied Biosystems | TaqMan miRNA assay, assay ID 000426 | – |
| hsa-let-7b miRNA assay kit | Primer | Applied Biosystems | TaqMan miRNA assay, assay ID 000378 | – |
| miR-191 miRNA assay kit | Primer | Applied Biosystems | TaqMan miRNA assay, assay ID 002299 | – |
| miR-103 miRNA assay kit | Primer | Applied Biosystems | TaqMan miRNA assay, assay ID 000439 | – |
| Penicillin-Streptomycin solution (100×) | Reagent | Sigma–Aldrich | P4333 | Original brand not specified |
| Insulin | Reagent | Sigma–Aldrich | I-6634 | – |
| FcR blocking reagent | Reagent | Miltenyi Biotec | 120-000-442 | – |
| FBS | Reagent | Sigma–Aldrich | F2442 | Replaces Benchmark Cat. #100-106 |
| 7-AAD | Reagent | Molecular Probes | A-1310 | – |
| M-MLV reverse transcriptase | Reagent | Invitrogen | AM2044 | – |
| Insulin | Reagent | Sigma–Aldrich | I-6634 | – |
| Bovine serum albumin (BSA) | Reagent | Sigma–Aldrich | 05470 | Original brand not specified |
| Platinum Pfx DNA polymerase | Reagents | Invitrogen | 11708013 | – |
| 10 mM sNTP mix | Reagent | Invitrogen | 18427-013 | Original brand not specified |
| Dulbecco's phosphate buffered saline (PBS); without $Ca^{2+}$ and $Mg^{2+}$ | Reagent | Sigma–Aldrich | D8537 | Original brand not specified |

## Procedure

1. CD44-based purification by FACS (volumes given below are for 5 million cells). Note: This step contains information described in *Patrawala and colleagues (2006)*, *(2007)* and *Li and colleagues (2009)*.
    a. Centrifuge the single cell suspension from Protocol 2 at 380×*g* for 5 min at 4°C and resuspend cells in 90 µl of cold staining buffer (PBS, pH 7.2, 0.5% BSA, 5 µg/ml insulin).
       i. Remove 10 µl from cell suspension and add to 90 µl staining buffer in another tube for isotype control staining.
    b. Add 20 µl of FcR blocking reagent to the remaining 80 µl of cell suspension and incubate at 4°C in the dark for 10 min.
    c. Add 10 µl of FITC-conjugated monoclonal anti-CD44 to the cell suspension, mix by tapping, and incubate at 4°C in the dark for 15 min.
       i. Note: antibody comes prediluted from the manufacturer and no additional dilution is necessary.
       ii. Gently tap every 5 min to prevent cells from settling down.
       iii. For isotype control use 10 µl FITC-mouse IgG2b isotype control immunoglobulin.
    d. Add 5 ml of cold PBS and spin at 380×*g* for 5 min at 4°C. Resuspend cells in cold serum-free IMDM with GLUTAMAX + 1% pen/strep at a concentration of $2 \times 10^6$ cells/ml in a 5 ml polystyrene tube at no more than 2.5 ml/tube.
       i. Keep on ice and shielded from light.
    e. Prepare collection tubes using 5 ml polypropylene tubes containing 1 ml of FBS + 1% pen/strep.
    f. Add 7-AAD to 1 µg/ml (stock = 100 µg/ml) to the sort tube 10 min before analysis.

g. Analyze and sort by FACS (Coulter Epics Elite flow cytometer), electronically gating out cell debris and cell aggregates. Collect only the top 10% of the most brightly stained cells for the positive population (Cohort 1: CD44+) and only the bottom 10% most dimly stained cells for the negative population (Cohort 2: CD44−).
    i. Re-run the isolated cells to confirm their purity.
2. qRT-PCR analysis. Note: This step contains information described in *Wiggins and colleagues (2010)*.
    a. Isolate total RNA from sorted cells.
        i. Measure RNA concentrations and record the $A_{260/280}$ and $A_{260/230}$ ratios using a NanoDrop 2000#.
    b. Heat-denature 10 ng of total RNA and miRNA-specific RT-primers at 70°C for 2 min and reverse transcribe using M-MLV reverse transcriptase following manufacturer's instructions.
    c. Determine gene expression levels by real-time PCR using Platinum *Pfx* DNA Polymerase and the Stratagene MX3005P Real-time PCR system#. All reactions should be optimized# and run in triplicate.
        i. miR-34a specific primers/probe.
        ii. hsa-let-7b specific primers/probe.
        iii. Internal references.
            • miR-103 specific primers/probe.
    d. Analyze:
        i. Define the threshold cycle (Ct) as the fractional cycle number at which fluorescence exceeds the fixed threshold of 0.2.
        ii. Analyze using the dCt (the Ct value normalized to miR-103 miRNA levels) and ddCt (difference between the dCt of marker positive population and that of the negative population) values for each of the miRNAs and markers.
        iii. Convert ddCt to percentage of expression using the formula $(2^{-ddCt})$.
3. Repeat experiment two more times.

## Deliverables

• Data to be collected:
    ○ FACS plots of CD44 purification.
    ○ Post-sort FACS analysis of CD44 purity.
    ○ Measured RNA concentrations after small RNA purification as well as $A_{260/280}$ and $A_{260/230}$ ratios.
    ○ Raw Ct qRT-PCR values, dCt, ddCT, and percentage of expression.
    ○ Graph of percentage of expression for each condition (Compare to Figure 1B.)

## Confirmatory analysis plan

• Statistical Analysis of the Replication Data:
    ○ At the time of analysis, we will perform the Shapiro–Wilk test and generate a quantile–quantile (q–q) plot to attempt to assess the normality of the data and also perform Levene's test to assess homoscedasiticity. If the data appears skewed, we will attempt a transformation in order to proceed with the proposed statistical analysis listed below and possibly perform the appropriate non-parametric test.
        ■ Compare percentage expression levels of miR-34a to levels of let-7b.
            • unpaired two-tailed Student's *t*-test.
• Meta-analysis of original and replication attempt effect sizes:
    ○ Because the original data is qualitative, a meta-analysis cannot be performed. The replication data will be plotted as means and 95% confidence intervals and the original data will be presented as a single point on the same plot for comparison.

## Known differences from the original study

• The replication will be restricted to LAPC4 cells only.
• The replicating lab will use a Stratagene MX3005P Real-time PCR system in place of the ABI Prism 7900 SDS used by the original authors.
• Based on the recommendation of the original authors, total RNA will be isolated in Step 2a instead of isolating only the small RNA fraction.

## Provisions for quality control

RNA sample purity ($A_{260/280}$ and $A_{260/230}$ ratios) will be reported for each sample. An isotype control will be used during the FACS experiment. All data obtained from the experiment—raw data, data analysis, control data and quality control data—will be made publicly available, either in the published manuscript or as an open access dataset available on the Open Science Framework (https://osf.io/gb7sr/).

## Protocol 4: Effects of miR-34a on tumor growth

This protocol describes how to inject LAPC4 tumor cells infected with a lentivirus containing miR-34a into NOD/SCID mice and measure the resultant tumor size, as seen in Supplemental Figure 5C.

### Sampling

- This experiment will be performed with 7 mice per group for a final power of 81.2%.
  - See Power Calculations section for details.
- The experiment consists of three cohorts of NOD/SCID mice:
  - Cohort 1: injected with uninfected LAPC4 cells (additional control).
    - N = 7.
      - Based on power calculations performed for the original two groups.
  - Cohort 2: injected with control-infected LAPC4 cells.
    - N = 7.
  - Cohort 3: injected with miR-34-infected LAPC4 cells.
    - N = 7.

### Materials and reagents

| Reagent | Type | Manufacturer | Catalog # | Comments |
|---|---|---|---|---|
| HEK293T cells | Cells | ATCC | #CRL-3216 | Replaces 293FT cells (Clontech, R700-07) originally used |
| HT1080 cells | Cells | ATCC | #CCL-121 | – |
| LAPC4 single cell suspension | Cells | From Protocol 2 | | |
| ultra-fine II insulin syringe, ½ cc, 31G needle | Material | BD | 328468 | – |
| 15 cm tissue culture dish | Materials | Corning (Sigma–Aldrich) | CLS430599 | Original brand not specified |
| 0.45 µm syringe filters | Materials | EMD Millipore | SLHP033RS | Original brand not specified |
| 6 well tissue culture plate | Materials | Corning (Sigma–Aldrich) | CLS3516 | Original brand not specified |
| IMDM with GlutaMAX, HEPES and sodium pyruvate | Medium | Gibco/Life Technologies | 31,980-030 | This reagent was referred to as 'DMEM' in the original study |
| IMDM with HEPES and sodium pyruvate | Medium | GIBCO/Life Technologies | 12440053 | – |
| EMEM | Medium | ATCC | 30-2003 | Original culture conditions for HT1080 cells unspecified |
| IMDM with GlutaMAX, HEPES and sodium pyruvate | Medium | Gibco/Life Technologies | 31,980-030 | This reagent was referred to as 'DMEM' in the original study |
| NOD/SCID mice, male, 6–8 weeks old | Mice | Jackson Laboratories | 001303 | – |
| REV packaging plasmid | Plasmid | Provided by original authors | | |
| RRE packaging plasmid | Plasmid | Provided by original authors | | |
| VSVG packaging plasmid | Plasmid | Provided by original authors | | |
| Lenti-ctl plasmid | Plasmid | System Biosciences | PMIRH000PA-1 | – |
| Lenti-miR-34 plasmid | Plasmid | System Biosciences | PMIRH34aPA-1 | – |
| GenElute endotoxin-free plasmid maxiprep kit | Reagent | Sigma–Aldrich | PLEX15-1KT | This kit replaces the Qiagen endo-free maxiprep kit used by the original authors |
| Lipofectamine 2000 | Reagent | Invitrogen/Life Technologies | 11,668-019 | – |
| FBS | Reagent | Sigma–Aldrich | F2442 | Replaces Benchmark Cat. #100-106 |
| Matrigel, phenol red-free | Reagent | Corning | 356237 | Same item as BD Cat. #CB-40234C |
| Penicillin-Streptomycin solution (100×) | Reagent | Sigma–Aldrich | P4333 | Original brand not specified |
| Fluorescent microscope | Equipment | Make and model to be recorded later | | Original unspecified |
| Ultra-low attachment T75 tissue culture flasks | Materials | Corning | 3814 | Original unspecified |
| Dissecting microscope | Equipment | Make and model to be recorded later | | Original unspecified |

## Procedure

Notes:

- HEK293 cells are maintained in IMDM with GLUTAMAX + 7% FBS without antibiotics.
- HT1080 cells are maintained in EMEM + 10% FBS with antibiotics.
  ○ All cells are maintained at 37°C with 5% $CO_2$.
- All cells will be sent for mycoplasma testing and STR profiling.

1. Generate lentivirus. Note: This step contains information described in *Jeter and colleagues (2009)*.
   a. Grow and prep all plasmids using Sigma Maxiprep kit according to manufacturer's instructions.
      i. Total DNA needed for one round of transfection:
         - RRE: 12 µg.
         - REV: 8 µg.
         - VSVG: 8 µg.
         - Lenti-ctl: 6 µg.
         - Lenti-miR-34a: 6 µg.
   b. Seed HEK293 packaging cells at 6 million cells per 15 cm dish for each transfection.
   c. The next day transfect with 6 µg RRE, 4 µg REV, and 4 µg VSVG packaging plasmids, along with the 6 µg of the lentiviral vector (lenti-ctl or lenti-miR-34a) using Lipofectamine 2000 at a 1:2 ratio of DNA (µg) to transfection reagent (µl) following manufacturer's instructions.
   d. After 36–48 hr, collect media containing virus and add fresh media.
      i. This initial collected media can be stored briefly at 4°C.
   e. After an additional 12–24 hr of culture, collect viral supernatants again and pool with first collection.
   f. Concentrate viral stock.
      i. Centrifuge the viral supernatant at 3000 rpm for 15 min to remove any cell debris.
      ii. Filter the supernatant through a 0.45 µm syringe filter.
      iii. Ultracentrifuge at 22,000 rpm for 2 hr at 4°C to produce concentrated viral stocks.
   g. Determine lentivirus titer for GFP using HT1080 cells.
      i. 1 day before harvesting viral supernatant, plate $1.2 \times 10^5$ HT1080 cells per well of a six well dish.
      ii. On the day of viral supernatant harvesting, count the number of cells in one well to determine cell number at time of infection.
      iii. Add a range of volumes between 2 and 5 µl of concentrated viral supernatant to the wells. Incubate for 72 hr.
      iv. Using a fluorescent microscope, count the number of GFP-positive cells per well; roughly 1–10% of cells will be GFP positive.
      v. Calculate the number of transfection units (TU/ml):
         - Divide the number of GFP-positive cells by the number of total cells counted.
         - Multiply that by the number of cells at the time of infection.
         - Multiply that by 1 divided by the final viral solution in µl multiplied by 1000.
         - This will yield the number of transfection units per ml.
2. Infect LAPC4 single cell suspension from Protocol 2 at a multiplicity of infection (MOI) of 10.
   a. *Keep LAPC4 cells at 37°C/5% $CO_2$ in IMDM with GLUTAMAX + 10% FBS in ultra-low attachment T75 flasks.
3. 24 hr post infection, harvest 10,000 cells per xenograft injection and harvest the remaining cells for qRT-PCR measurement.
   a. At time of harvest, visually examine the percentage of transduced cells by assessing GFP expression using a fluorescent microscope; ensure that greater than 90% of cells are GFP positive.
      i. Perform cell counts and record percentage of transduced cells. If percentage is less than 90%, discard cells and repeat transduction to obtain a 90% transduced population.
   b. 210,000 cells total for 21 injections.
      i. 7 injections per group.

- 70,000 uninfected LAPC4 cells.
- 70,000 lenti-ctl infected LAPC4 cells.
- 70,000 miR-34a infected LAPC4 cells.

 c. Snap freeze remaining cells for RNA extraction and store at −80°C until use in Protocol 6.

 i. Snap-freeze an equal number of uninfected, lenti-ctl infected and miR-34a infected LAPC4 cells.

 ii. Split each batch of cells into three aliquots as technical replicates for Protocol 6.

4. Subcutaneous (s.c.) injection of LAPC4 cells. Note: This step contains information described in *Patrawala and colleagues (2005)*, (*2006*), (*2007*) and *Li and colleagues (2009)*.

 a. For each injection, centrifuge 10,000 cells at 380×*g* for 5 min at RT and resuspend in 20 µl culture medium (IMDM + 15% FBS).

 b. Mix with 20 µl Matrigel on ice.

 c. Hold a 6–8 week old male NOD/SCID mice and inject the cell/Matrigel mixture into the flank by using an ultra-fine II insulin syringe, ½ cc, 31G needle.

 d. Repeat for remaining mice.

5. Monitor tumor development starting from the second week after injection.

 a. Measure tumor incidence (numbers of tumors/number of injections).

 b. Measure latency (time from injection to detection of palpable tumors) (additional parameter).

 c. Measure tumors weekly with caliper measurements (height, width, and depth to determine volume) (additional parameter).

6. Terminate all animals when the average weight of the control group tumors reaches 0.54 g; in the original study, this was at the 69 day mark. Dissect out tumors, and image.

 a. To convert caliper measurements to approximate weight, use data on tumor volume and tumor weight from Protocol 1 to calculate the mean density of the control group tumors. Use this density to calculate weight of the current tumors from measured volume.

 i. Only control group tumors will be analyzed at this stage to assess when to terminate all animals in the experiment.

 b. After dissection, weigh tumors.

 c. Image whole tumors by brightfield imaging.

 i. Please include a ruler to provide a size scale.

 d. Divide tumor into two pieces: one for protein extraction (Protocol 5) and one for RNA extraction (Protocol 6).

 i. Tumor tissue can be snap frozen for RNA and protein extraction.

## Deliverables

- Data to be collected:
  - Record of LAPC4 xenograft tumor maintenance (tumor incidence, latency [additional parameter], tumor caliper measurements [and volume (additional parameter)], tumor weight at harvest, time of harvest) (Compare to S5C).
  - Images of tumors by brightfield microscopy (Compare to Figure 5C).
- Sample delivered for further analysis:
  - Tumor tissue processed for protein extraction (Protocol 5) and RNA extraction (Protocol 6).
  - Snap frozen cells for RNA extraction (Protocol 6).

## Confirmatory analysis plan

- Statistical Analysis of the Replication Data:
  - At the time of analysis, we will perform the Shapiro–Wilk test and generate a quantile–quantile (q–q) plot to attempt to assess the normality of the data and also perform Levene's test to assess homoscedasiticity. If the data appears skewed, we will attempt a transformation in order to proceed with the proposed statistical analysis listed below and possibly perform the appropriate non-parametric test.
    - Compare mean tumor weight in miR-34a expressed tumors relative to lenti-ctl control tumors. Also compare to uninfected control tumors (additional control).
      - One-way ANOVA followed by planned comparisons using Fisher's LSD.

- ○ Lenti-control vs miR-34a.
- ○ Uninfected vs miR-34a.
  - ■ Compare tumor incidence in miR-34a expressed tumors relative to lenti-ctl tumors. Also compare to uninfected control tumors (additional control).
    - • One-way ANOVA followed by planned comparisons using Fisher's LSD.
      - ○ Lenti-control vs miR-34a.
      - ○ Uninfected vs miR-34a.
  - ■ Compare tumor latency in miR-34a expressed tumors relative to lenti-ctl tumors. Also compare to uninfected control tumors (additional control).
    - • Fisher's Exact Test.
  - ■ Compare tumor volume over time in miR-34a expressed tumors relative to lenti-ctl tumors. Also compare to uninfected control tumors (additional control).
    - • Calculate area under the curve (AUC) for each mouse and perform a one-way ANOVA followed by planned comparisons using Fisher's LSD.
    - • Lenti-control vs miR-34a.
    - • Uninfected vs miR-34a.
- • Meta-analysis of original and replication attempt effect sizes:
  - ○ This replication attempt will perform the statistical analysis listed above, compute the effects sizes, compare them against the reported effect size in the original paper and use a meta-analytic approach to combine the original and replication effects, which will be presented as a forest plot.

## Known differences from the original study

### Provisions for quality control

All data obtained from the experiment—raw data, data analysis, control data and quality control data—will be made publicly available, either in the published manuscript or as an open access dataset available on the Open Science Framework (https://osf.io/gb7sr/).

- • We are confirming that >90% of injected LAPC4 cells have been successfully transduced with the vector.
- • We are recording tumor latency and tumor volume as additional parameters.
- • Only control tumors' weight will be used to determine the experimental endpoint for tumor growth.

## Protocol 5: Western blot analysis of miR34a infected LAPC4 tumor tissue (additional related experiment)

This protocol describes Western blot analysis of CD44 levels from tumor tissue derived from Protocol 4. In the original study, this was performed in Figure 4A (right panels) on tumor tissues derived from DU145 and PC3 xenograft tumors. This experiment, however, will be performed on LAPC4 xenograft tumor tissue to maintain consistency of tumor materials throughout the replication plan and is thus exploratory in nature.

### Sampling

- • This protocol will use the samples derived from Protocol 4.
  - ○ Original data presented is qualitative; representative images. The replication attempt is exploratory in nature and not looking for a specific effect.
- • The experiment consists of three cohorts:
  - ○ Cohort 1: tumors from uninfected LAPC4 cells.
  - ○ Cohort 2: tumors from control-infected LAPC4 cells.
  - ○ Cohort 3: tumors from miR-34-infected LAPC4 cells.
  - ○ Each cohort is probed by Western blot for:
    - ■ Anti-CD44.
    - ■ Anti-beta-actin.

## Materials and reagents

| Reagent | Type | Manufacturer | Catalog # | Comments |
|---|---|---|---|---|
| Rabbit monoclonal α CD44 antibody | Antibody | AbCam | ab51037 | – |
| Mouse monoclonal IgG2_b α beta-actin | Antibody | Cell Signaling Technology | 3700S | Original brand not specified |
| IRDye 800CW goat anti-rabbit IgG(H + L) | Antibody | LiCor | 926-32211 | – |
| IRDye 680LT goat anti-mouse IgG(H + L) | Antibody | LiCor | 926-68020 | – |
| BCA protein assay | Kit | Pierce | 23227 | – |
| PVDF membrane | Material | Millipore | IPVH00010 | – |
| NuPAGE™ 4–12% Bis-Tris gels 1.0 mm, 10 well | Material | Invitrogen | NP0005 | – |
| RIPA buffer | Reagent | Pierce | 89,901 | – |
| Phosphatase inhibitor cocktail II | Reagent | Sigma–Aldrich | P5726 | – |
| Phosphatase inhibitor cocktail III | Reagent | Sigma–Aldrich | P0044 | – |
| Complete-EDTA free tablet | Reagent | Roche | 05892791001 | – |
| NuPAGE™ MOPS SDS running buffer (20×) | Reagent | Invitrogen | NP0005 | – |
| NuPAGE™ LDS sample buffer (4×) | Reagent | Invitrogen | NP0007 | – |
| Sample reducing agent | Reagent | Invitrogen | NP0009 | – |
| NuPAGE transfer buffer (20×) | Reagent | Invitrogen | NP006-1 | – |
| Prestained protein marker | Reagent | Fermentas | SM0671 | – |
| Ethanol | Reagent | Sinopharm Chemical Reagent | 10009218 | – |
| Sodium chloride (NaCl) | Reagent | Sinopharm Chemical Reagent | 10019318 | – |
| Tween-20 | Reagent | Sigma–Aldrich | P1379 | – |
| Tris buffered saline (TBS); 10× solution | Reagent | Sigma–Aldrich | T5912 | Original brand not specified |
| Non-fat powder milk | Reagent | Sangon | NB0669-250g | – |
| Antibody dilute reagent solution | Reagent | Invitrogen | 00-3218 | – |

### Procedure

Note: at authors' recommendation, the replicating lab will use their standard western blot protocol.

1. Make a total protein lysate from tumor tissue from Protocol 4.
    a. Transfer a piece of tumor sample into a 5 ml polypropylene tube containing 800–1000 μl lysis buffer.
        i. Lysis buffer: Add 1 tablet of complete-EDTA free tablet, 100 μl Phosphatase Inhibitor Cocktail II and II to 10 ml RIPA buffer.
    b. Homogenize the tumor sample on ice with setting on 6 (equal to 30,000 rpm) for 5 s on and 5 s off until it is homogenized completely.
    c. Transfer the lysate into a 1.5 ml tube and sonicate on ice for 15 s.
    d. Spin at 13,000 rpm for 20 min at 4℃ to remove tissue debris.
    e. Spin the resultant supernatant at 13,000 rpm for 20 min at 4℃.
    f. Aliquot the supernatant and store at −80℃.
2. Quantify total protein concentration.
    a. Prepare BCA standard (0, 25 μg/ml, 125 μg/ml, 250 μg/ml, 500 μg/ml, 750 μg/ml, 1000 μg/ml, 1500 μg/ml, 2000 μg/ml) according to the user manual.
    b. Dilute tumor lysate sample with PBS at 1:10 ratio.
    c. Pipette 25 μl of each standard or sample duplicate into 96-well plate.
    d. Add 200 μl of the working reagent (50:1, Reagent A:B) to each well and mix plate thoroughly on a plate shaker for 30 s.
    e. Cover plate and incubate at 37℃ for 30 min.

 f. Cool plate to RT. Measure the absorbance at or near 562 nm on a plate reader. Calculate total protein amount of each sample.

3. Preparation of sample:
 a. Mix tumor lysate with 4× LDS sample buffer plus reducing agent. Boil at 70°C for 5 min. Cool down on ice before centrifuge briefly.

4. Preparation of gel:
 a. Unpack NuPAGE 4–12% Bis-Tris gel and wash wells with running buffer. Assemble the gel into the cassette. Fill the inner and outer chamber with running buffer.

5. Running the gel:
 a. Load samples. Run with constant voltage (150–200 V). Stop running when the dye is close to the lower edge, about 1.5 hr.

6. Membrane transfer:
 a. Pre-activate PVDF membrane in ethanol or methanol for 1 or 2 min, followed by pre-wetting it and filter papers in cold transfer buffer.
 b. Assemble 'sandwich' for Bio-Rad's semi-dry transblot. Layer from bottom to top: filter paper—membrane—gel—filter paper.
 c. Semi-dry transfer for 90 min at 25 V.

7. When the transfer has finished, immerse PVDF membrane in 5% non-fat milk TBS/0.1%T solution and gently shake for 1 hr at RT.

8. Incubate with primary antibody diluted in antibody dilute solution at 4°C overnight.
 a. Anti-CD44: 1:1000.
 b. Anti-beta-actin: 1:5000.
 i. Loading control.

9. Wash membrane in TBS with 0.1% Tween-20, 5 min × 3.

10. Add IRDye 800CW/680LT Goat anti-rabbit/mouse IgG (diluted 1:5000 with 0.5% non-fat milk TBS/0.1%T solution) and incubate 1 hr at RT, protected from light.

11. Wash membrane in TBS/0.1%T, 5 min × 3, protected from light. Soak in TBS to remove Tween-20.

12. Detect with Odyssey at 800 and 700 channel.

## Deliverables

- Data to be collected:
  - Full images, including ladder, of western blots for CD44 and beta-actin for each tumor sample.
  - Calculate band intensity for each band as ratio to loading control band.

## Exploratory analysis plan

- Statistical Analysis of the Replication Data:
  - At the time of analysis, we will perform the Shapiro–Wilk test and generate a quantile–quantile (q–q) plot to attempt to assess the normality of the data and also perform Levene's test to assess homoscedasiticity. If the data appears skewed, we will attempt a transformation in order to proceed with the proposed statistical analysis listed below and possibly perform the appropriate non-parametric test.
    - Compare CD44 protein across all three conditions.
      - One way ANOVA followed by planned comparisons using Fisher's LSD:
        - CD44 protein levels in miR-34a expressed tumors vs to lenti-ctl tumors.
        - CD44 protein levels in miR-34a expressed tumors vs to uninfected control tumors (additional control).
- Meta-analysis of original and replication attempt effect sizes:
  - Meta-analysis will not be performed as this experiment is exploratory in nature and there is no matching original data with which to statistically compare the replication data.

## Known differences from the original study

- While the original study performed Western blot analysis on DU145 and PC3 xenograft tumor tissue, this experiment will perform the same analysis on LAPC4 xenograft tumor tissue.
- Uninjected LAPC4 cells and tumors derived from uninjected LAPC4 cells have been added as an additional control.
- Because of these changes, the experiment is not directly confirmatory in nature, but rather exploratory. Thus, we will not directly power the experiment, but we have determined our sensitivity based on the number of samples we will use.

## Provisions for quality control

All data obtained from the experiment—raw data, data analysis, control data and quality control data—will be made publicly available, either in the published manuscript or as an open access dataset available on the Open Science Framework (https://osf.io/gb7sr/).

## Protocol 6: qRT-PCR analysis of miR34 infected LAPC4 cells and tumor tissue

This protocol describes how to assess miR-34a levels in tumor tissue derived from lenti-miR-34a infected LAPC4 cells. This is an additional experiment added by the RP:CB core team and is partially based on Supplemental Figure 4A. It is a quality control check to ensure that increased levels of miR-34a are still present even after the infected LAPC4 cells have been injected into mice to form tumors.

### Sampling

- This protocol will use samples derived from Protocol 4.
  ○ Original data presented is qualitative; representative images. The replication attempt is exploratory in nature and not looking for a specific effect.
- The experiment consists of six cohorts:
  ○ RNA derived from uninfected LAPC4 cells.
  ○ RNA derived from lenti-ctl LAPC4 cells.
  ○ RNA derived from lenti-miR-34a infected LAPC4 cells.
    ■ Each cohort of cells has three technical replicates.
  ○ RNA derived from uninfected LAPC4 xenograft tumor tissue.
  ○ RNA derived from lenti-ctl LAPC4 xenograft tumor tissue.
  ○ RNA derived from lenti-miR-34a infected LAPC4 xenograft tumor tissue.
    ■ Each cohort of tumors has seven biological replicates.
  ○ In each cohort, qRT-PCR will be performed for:
    ■ miR-34a.
    ■ miR-103 (housekeeping miRNA, additional control).

### Materials and reagents

| Reagent | Type | Manufacturer | Catalog # | Comments |
|---|---|---|---|---|
| miR-34a miRNA assay kit | Primer | Applied Biosystems | TaqMan miRNA assay, assay ID 000426 | – |
| miR-191 miRNA assay kit | Primer | Applied Biosystems | TaqMan miRNA assay, assay ID 002299 | – |
| miR-103 miRNA assay kit | Primer | Applied Biosystems | TaqMan miRNA assay, assay ID 000439 | – |
| M-MLV reverse transcriptase | Reagent | Invitrogen | AM2043 | – |
| Platinum Pfx DNA polymerase | Reagents | Invitrogen | 11708013 | – |
| MX3005P Real-time PCR system | Equipment | Stratagene | – | Replaces ABI prism 7900 SDS |
| 10 mM sNTP mix | Reagent | Invitrogen | 18427-013 | Original brand not specified |

### Procedure

Note: This procedure contains information described in *Wiggins and colleagues (2010)*.

1. Isolate total RNA from frozen cells and tumor tissue from Protocol 4.
   a. Measure and record RNA concentrations as well as $A_{260/280}$ and $A_{260/230}$ ratios.
2. Heat denature 10 ng of total RNA and miRNA-specific RT-primers at 70°C for 2 min and reverse transcribe using M-MLV reverse transcriptase following manufacturer's instructions.
3. Determine gene expression levels by real-time PCR using Platinum *Pfx* DNA Polymerase and the Stratagene MX3005P Real-time PCR system[#]. All reactions should be optimized[#] and run in triplicate.
   i. miR-34a specific RT primer.
   ii. Internal references.
      a. miR-103 specific RT primer.
4. Analyze qPCR raw data.

   a. Define the threshold cycle (Ct) as the fractional cycle number at which fluorescence exceeds the fixed threshold of 0.2.
   b. Analyze using the dCt (the Ct value normalized to miR-103 levels) values for each of the miRNAs and markers.

### Deliverables

- Data to be collected:
  ○ Measured RNA concentrations after small RNA purification, including the $A_{260/280}$ and $A_{260/230}$ ratios.
  ○ Raw Ct qRT-PCR values and calculated dCt.
  ○ Graph of miR-34a levels (dCt) for each condition.

### Exploratory analysis plan

- Statistical Analysis of the Replication Data:
  ○ At the time of analysis, we will perform the Shapiro–Wilk test and generate a quantile–quantile (q–q) plot to attempt to assess the normality of the data and also perform Levene's test to assess homoscedasiticity. If the data appears skewed, we will attempt a transformation in order to proceed with the proposed statistical analysis listed below and possibly perform the appropriate non-parametric test.
    ■ One way ANOVA of miR-34a levels across all three conditions in cells before injection followed by planned comparisons using Fisher's LSD:
      • Lenti-clt control cells compared to miR-34a expressed cells.
      • Uninfected cells compared to miR-34a expressed cells.
    ■ One way ANOVA of miR-34a levels across all three conditions in tumors followed by planned comparisons using Fisher's LSD:
      • Lenti-clt control tumors compared to miR-34a expressed tumors.
      • Uninfected tumors compared to miR-34a expressed tumors.
- Meta-analysis of original and replication attempt effect sizes:
  ○ Meta-analysis will not be performed as this experiment is exploratory in nature and there is no matching original data with which to statistically compare the replication data.

### Known differences from the original study

- The replication will be restricted to LAPC4 cells.
- We will run the qRT-PCR analysis on tumors derived from the treated LAPC4 cells as well as on the infected LAPC4 cells directly.
- Uninjected LAPC4 cells and tumors derived from uninjected LAPC4 cells have been added as an additional control.
- Because of these changes, the experiment is not directly confirmatory in nature, but rather exploratory. Thus, we will not directly power the experiment, but we have determined our sensitivity based on the number of samples we will use.

### Provisions for quality control

RNA sample purity ($A_{260/280}$ and $A_{260/230}$ ratios) will be reported for each sample. All data obtained from the experiment—raw data, data analysis, control data and quality control data—will be made publicly available, either in the published manuscript or as an open access dataset available on the Open Science Framework (https://osf.io/gb7sr/).

### Protocol 7: Luciferase assays confirming binding of miR-34a to putative binding sites in the 3′ UTR of CD44

This protocol describes how to perform luciferase assays in order to demonstrate that binding of miR-34a to its binding sites in the 3′ UTR of CD44 decreases luciferase activity. It also tests whether this decrease is abrogated when the seed regions of the putative miR-34a binding sites are mutated, as seen in Figure 4D. The replication will use only the pMIR-CD443′ UTR and pMIR-CD44M1M23′ UTR constructs.

## Sampling

- The experiment will be repeated a total of 16 times for a final power of 80.4%.
- Each experiment consists of two cohorts;
  - Cohort 1: DU145 cells transfected with pMIR-CD443′ UTR.
  - Cohort 2: DU145 cells transfected with pMIR-CD44M1M23′ UTR.
- Each cohort is treated with miR-34a or a non-coding control and levels of luciferase activity are measured.

## Materials and reagents

| Reagent | Type | Manufacturer | Catalog # | Comments |
|---|---|---|---|---|
| pMIR-CD443′ UTR | Plasmid | Provided by the original author | | Vector containing the 3′ UTR of CD44 fused to luciferase |
| pMIR-CD44M1M23′ UTR | Plasmid | Provided by the original author | | Vector containing the 3′ UTR of CD44 fused to luciferase, with both seed regions mutated |
| phRL-CMV | Plasmid | Provided by the original author | | Renilla control |
| DU145 cells | Cells | ATCC | HTB-81 | – |
| Dual luciferase assay | Kit | Promega | E1960 | – |
| miR34-a oligo | Oligo | Synthesis left to the discretion of the replicating lab | | – |
| Non-coding control oligo | Oligo | Synthesis left to the discretion of the replicating lab | | – |
| RPMI 1640 | Media | Thermal Scientific | SH30809 | – |
| Fetal bovine serum (FBS) | Reagent | Gibco | 10099-133 | – |
| Lipofectamine 2000 | Reagent | Life Technologies | 11668027 | – |
| Fluorescence plate reader | Equipment | PerkinElmer | Envision 2104 multilabel reader | Replaces Gen-Probe Leader 50i lunimometer used originally |
| Endo-free maxiprep kit | Kit | Qiagen | 12362 | Original unspecified |
| miR-34a mirVana mRNA mimic | Oligo | Ambion/Life Technologies | 4464066 | Termed the miR34a oligo |
| mirVana miRNA mimic, miR-1 positive control | Oligo | Ambion/Life Technologies | 464062 | Termed the NC oligo |

## Procedure

Notes:

- DU145 cells are maintained in RPMI1640 supplemented with 10% FBS.
- All cells will be sent for STR profiling and mycoplasma testing.

1. Transform, grow and maxiprep the following plasmids using an endo-free Maxiprep kit according to the manufacturer's instructions.
   a. pMIR-CD443′ UTR.
   b. pMIR-CD44M1M23′ UTR.
   c. phRL-CMV.
2. Sequence inserts to confirm vector identity and run on an agarose gel to confirm vector integrity.
3. Seed $3 \times 10^4$ DU145 cells per well in 24-well plates (27 wells total).
   a. Let grow overnight.
4. 24 hr after plating, transfect cells with pMIR-3′ UTR or pMIR-CD44mut3′ UTR, miR-34a oligo or NC oligo and phRL-CMV using Lipofectamine 2000 according to the manufacturer's instructions.
   a. 1 µg pMIR-CD443′ UTR + 24 pmoles miR-34a oligo + 1 ng phRL-CMV.
   b. 1 µg pMIR-CD443′ UTR + 24 pmoles NC oligo + 1 ng phRL-CMV.
   c. 1 µg pMIR-CD44M1M23′ UTR + 24 pmoles miR-34a oligo + 1 ng phRL-CMV.
   d. 1 µg pMIR-CD44M1M23′ UTR + 24 pmoles NC oligo + 1 ng phRL-CMV.
   e. Untransfected cells.
   f. Empty wells (media only).
   g. Each condition should be run in quadruplicate (four wells).

5. 48 hr later, read the ratio of luciferase to renilla using the Promega dual luciferase assay according to the manufacturer's instructions.
6. Repeat experiment from Step 3 onwards an additional 15 times.

## Deliverables

- Data to be collected:
  - ○ Sequence data confirming vector identity.
  - ○ Whole gel images of agarose gels confirming vector integrity.
  - ○ Raw luciferase and renilla readings for each well.
  - ○ Normalized renilla and luciferase readings based on empty well and cells-only wells.
  - ○ Ratio of luciferase to renilla averaged across the four wells of each condition.

## Confirmatory analysis plan

- At the time of analysis, we will perform the Shapiro–Wilk test and generate a quantile–quantile (q–q) plot to attempt to assess the normality of the data and also perform Levene's test to assess homoscedasiticity. If the data appears skewed, we will attempt a transformation in order to proceed with the proposed statistical analysis listed below and possibly perform the appropriate non-parametric test.
  - ○ Statistical Analysis of the Replication Data:
    - ■ Two way ANOVA (2 × 2 factorial) followed by Bonferroni corrected pairwise comparisons:
      - • Wt+NC vs wt+34a.
- Meta-analysis of original and replication attempt effect sizes:
  - ○ This replication attempt will perform the statistical analysis listed above, compute the effects sizes, compare them against the reported effect size in the original paper and use a meta-analytic approach to combine the original and replication effects, which will be presented as a forest plot.

## Known differences from the original study

- The replication attempt will use only the wild-type CD44 3′ UTR anad the 3′ UTR with both miR-34a seed regions mutated; it will exclude the two constructs containing one mutated seed region.

## Provisions for quality control

All data obtained from the experiment—raw data, data analysis, control data and quality control data—will be made publicly available, either in the published manuscript or as an open access dataset available on the Open Science Framework (https://osf.io/gb7sr/).

- STR profiling and mycoplasma testing results.
- Sequencing results and agarose gel images confirming vector integrity.

# Power calculations

## Protocol 1

- No power calculations required.

## Protocol 2

- No power calculations required.

## Protocol 3

### Summary of original data

- Original data is qualitative; only one experiment is shown without any error bars.
- Values were estimated from graph.

- Since no variance or sample size data were provided, power calculations will be performed with a range of variances and an assumed N per group of 3.

**Figure 1B**

| | Mean | Assumed variance | | | | |
|---|---|---|---|---|---|---|
| | | 2% | 15% | 28% | 40% | Assumed N |
| Let-7b levels in LAPC4 CD44$^+$ cells | 30% | 0.6 | 4.5 | 8.4 | 12 | 3 |
| miR-34 levels in LAPC4 CD44$^+$ cells | 3% | 0.06 | 0.45 | 0.84 | 1.2 | 3 |

## Test family

- Unpaired two-tailed Student's *t*-test, alpha error = 0.05.

## Power calculations

- Note: Calculations were performed using G*Power software (version 3.1.7) (*Faul et al., 2007*).

| Variance level | Group 1 vs | Group 2 | Effect size | A priori power | Group 1 N | Group 2 N |
|---|---|---|---|---|---|---|
| 2% | let-7B | miR-34a | 63.32377903 | >99.99% | 2* | 2* |
| 15% | let-7B | miR-34a | 8.443170537 | 97.06% | 2* | 2* |
| 28% | let-7B | miR-34a | 4.523127073 | 98.01% | 3 | 3 |
| 40% | let-7B | miR-34a | 3.166188951 | 82.2% | 3 | 3 |

*With three samples per group, achieved power is >99.99%.

In order to produce quantitative replication data, we will run the replication experiment three times. We will then determine the replication attempt's mean and variance from those three replicates. We will combine the replication variance with the original data, whose single datapoint we are assuming represents a mean. Combining the original mean and the replication variance will generate a simulated effect size. Using this simulated effect size, we will then determine the number of replicates required to reach 80% power and will perform additional replicates if necessary to ensure the replication is powered to at least 80%.

## Protocol 4
### Summary of original data

- Note: Original values presented in Supplemental Figure 5C. Authors confirmed tumor incidence was 100%.

| Supp. Figure 5C: Tumor weight in g | Mean | SD | N |
|---|---|---|---|
| NOD/SCID mice injected with LAPC4 cells infected with lenti-ctl | 0.54 | 0.32 | 6 |
| NOD/SCID mice injected with LAPC4 cells infected with lenti-miR-34 | 0.15 | 0.09 | 6 |

## Test family

- One-way ANOVA followed by planned comparisons using Fisher's LSD.

## Power calculations

- Calculations were performed using R software, version 3.1.2 (R Core Team, 2014) and G*Power software (version 3.1.7) (*Faul et al., 2007*).

**ANOVA, alpha error = 0.05**

| F (2,15) | $\eta_P^2$ | Effect size *f* | Power | Total sample size across all groups |
|---|---|---|---|---|
| 4.2865 | 0.363680 | 0.7560002 | 81.99% | 21 |

**Planned comparisons by Fisher's LSD, alpha error = 0.05**

| Group 1 vs | Group 2 | Effect size | A priori power | Group 1 sample size | Group 2 sample size |
|---|---|---|---|---|---|
| Lenti-Ctl | Lenti-miR-34 | 1.659199 | 81.2% | 7 | 7 |
| Uninfected | Lenti-miR-34 | 1.659199 | 81.2% | 7 | 7 |

- Note: the additional control group of uninfected cells was assumed to have the same mean and variance as the lenti-control group for calculation purposes.

# Protocol 5

- This experiment differs from the original it is based on, and is thus exploratory in nature. The number of sample is derived from Protocol 4.
- Based on the sample size from Protocol 4, with α of 0.05, we will be powered to 80% to detect an effect size *f* of 0.7379139 (ANOVA: Fixed effects, omnibus, one-way) and with α of 0.05, we will be powered to 80% to detect an effect size *d* of 1.6317141 (unpaired two-tailed *t*-test).

# Protocol 6

- This experiment differs from the original it is based on, and is thus exploratory in nature. The number of samples is derived from Protocol 4.

## Sensitivity calculations

- Calculations were performed using R software, version 3.1.2 (R Core Team, 2014) and G*Power software (version 3.1.7) (*Faul et al., 2007*).
- Based on the sample size from Protocol 4, with α of 0.05, we will be powered to 80% to detect an effect size *f* of 0.7379139 (ANOVA: Fixed effects, omnibus, one-way) and with α of 0.05, we will be powered to 80% to detect an effect size *d* of 1.6317141 (unpaired two-tailed *t*-test) for tumors.
- Based on the sample size from Protocol 4, with α of 0.05, we will be powered to 80% to detect an effect size *f* of 1.3573433 (ANOVA: Fixed effects, omnibus, one-way) and with α of 0.05, we will be powered to 80% to detect an effect size *d* of 3.0708923 (unpaired two-tailed *t*-test) for tumors.

# Protocol 7
## Summary of original data

- Note: data provided by original authors.

| Figure 4D | Luc activity | Sem | SD | N |
|---|---|---|---|---|
| Wt+NC | 1 | 0 | 0 | 3 |
| Wt+34a | 0.6708 | 0.087 | 0.15068842 | 3 |
| M1M2+NC | 1 | 0 | 0 | 3 |
| M1M2+34a | 0.7803 | 0.06 | 0.103923048 | 3 |

## Test family

- Two-way ANOVA (2 × 2 factorial) followed by Bonferroni corrected pairwise comparisons:
  ○ Wt+NC vs wt+34a.

## Power calculations

- Calculations were performed using GraphPad PRISM v6 for Mac and G*Power software (version 3.1.7) (*Faul et al., 2007*).

**ANOVA; $\alpha = 0.05$**

| F (1,8) | $\eta_P^2$ | Effect size $f$ | Power | Total sample size across all groups |
|---|---|---|---|---|
| 1.0740 | 0.118360 | 0.3664012 | 80.33% | 61 |

**T tests; $\alpha = 0.05$**

| Group 1 vs | Group 2 | Effect size | A Priori power | Group 1 sample size | Group 2 sample size |
|---|---|---|---|---|---|
| Wt+NC | Wt+34a | 3.08955 | 80.44% | 3 | 3 |

# Acknowledgements

The Reproducibility Project: Cancer Biology core team would like to thank the original authors, in particular Dr. Can (Julie) Liu and Dr. Dean Tang, for generously sharing critical information as well as reagents to ensure the fidelity and quality of this replication attempt. We thank Courtney Soderberg at the Center for Open Science for assistance with statistical analyses. We would also like to thanks the following companies for generously donating reagents to the Reproducibility Project: Cancer Biology; American Type and Tissue Collection (ATCC), Applied Biological Materials, BioLegend, Charles River Laboratories, Corning Incorporated, DDC Medical, EMD Millipore, Harlan Laboratories, LI-COR Biosciences, Mirus Bio, Novus Biologicals, Sigma–Aldrich, and System Biosciences (SBI).

# Additional information

### Group author details

**Reproducibility Project: Cancer Biology**

Elizabeth Iorns: Science Exchange, Palo Alto, California; William Gunn: Mendeley, London, United Kingdom; Fraser Tan: Science Exchange, Palo Alto, California; Joelle Lomax: Science Exchange, Palo Alto, California; Timothy Errington: Center for Open Science, Charlottesville, Virginia

### Competing interests

RP:CB: We disclose that EI, FT, and JL are employed by and hold shares in Science Exchange Inc. The experiments presented in this manuscript will be conducted by JL at Crown Bioscience, which is a Science Exchange lab. The other authors declare that no competing interests exist.

### Funding

| Funder | Author |
|---|---|
| Laura and John Arnold Foundation | Reproducibility Project: Cancer Biology |

The Reproducibility Project: Cancer Biology is funded by the Laura and John Arnold Foundation, provided to the Center for Open Science in collaboration with Science Exchange. The funder had no role in study design or the decision to submit the work for publication.

### Author contributions

JL, Drafting or revising the article; ML, Conception and design; RP:CB, Conception and design, Drafting or revising the article

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
