## [Decision Letter]

Thank you for sending your work entitled “Registered report: The microRNA miR-34a inhibits prostate cancer stem cells and metastasis by directly repressing CD44” for consideration at *eLife*. Your article has been favorably evaluated by Stylianos Antonarakis (Senior editor), a Reviewing editor, and five reviewers.

The following individuals responsible for the peer review of your submission have agreed to reveal their identity: Andrea Ventura and Julie Liu (peer reviewers). Three other reviewers remain anonymous.

The Reviewing editor and the reviewers discussed their comments before we reached this decision, and the Reviewing editor has assembled the following comments to help you prepare a revised submission.

1) The first author of the original study felt that the authors of this report understand the central hypothesis of the original study, and designed/chose three major and critical experiments to support the central hypothesis. However the referee expressed concern about the rationale for choosing LAPC4 as their only model to perform the replication experiments. Using an additional model like DU145, as was done in the original study, would be important.

2) Much of the confusion about miR-34a functions stem from the drastic differences between the gain-of-function phenotypes and the loss of function phenotypes. Overexpression of miR-34a invariably gives rise to strong tumor suppressive effects in multiple cell lines, while miR-34a null in mice yield very little tumor suppressor effects in vivo. Surprisingly, a number of studies, including the one reported by Liu et al., show a strong effect of anti-34a. Thus, two of the referees would like to request experiments to reproduce some of the anti-34a experiments reported in the original study, such as Figure 1J. These experiments were central to the conclusions drawn by Liu et al:

2A) Systemic miR-34 delivery inhibits prostate cancer metastasis and extends survival (point made in Abstract and results depicted in main Figure 3B, 3D). This is a key experiment that is central to the conclusion, especially given that this work is published in Nature Medicine with an emphasis on translational relevance.

2B) Anti-miR34 antagomir increases metastasis relative to a control antagomir (Figure 1J). This is central to the claim that endogenous miR-34a antagonizes prostate cancer metastasis.

2C) CD44 is a direct target f miR-34a in prostate cancer (Figure 4D). The authors show that miR-34a reduces luciferase activity in a reporter assay and that mutagenesis of the miRNA complementary sites abolish this repression. This is another key conclusion of the paper and should be replicated.

3) Given that the sample sizes can be as small as 2 or 3 (for example, protocol 3), I suggest the investigators to pursue exploratory examination of data and possibly non-parametric comparison methods, in addition to the proposed ANOVA methods, at the time of data analysis.

---

## [Author Response]

*1) The first author of the original study felt that the authors of this report understand the central hypothesis of the original study, and designed/chose three major and critical experiments to support the central hypothesis. However the referee expressed concern about the rationale for choosing LAPC4 as their only model to perform the replication experiments. Using an additional model like DU145, as was done in the original study, would be important*.

We agree that all of the experiments included in the original study are important, and choosing which experiments to replicate has been one of the great challenges of this project. The Reproducibility Project: Cancer Biology (RP:CB) aims to replicate experiments that are impactful, but does not necessarily aim to replicate all the impactful experiments in any given paper. In this case, the RP:CB core team felt that the most impactful information in [11] was performed in the LAPC4 model. We also agree the DU145 model, as well as the LAPC9 model, were important in contributing to the overall conclusions, however we are attempting to identify a balance of breadth of sampling for general inference with sensible investment of resources on replication projects to determine to what extent the included experiments are reproducible. As such, we will restrict our analysis to the experiments being replicated and will not include discussion of experiments not being replicated in this study.

*2) Much of the confusion about miR-34a functions stem from the drastic differences between the gain-of-function phenotypes and the loss of function phenotypes. Overexpression of miR-34a invariably gives rise to strong tumor suppressive effects in multiple cell lines, while miR-34a null in mice yield very little tumor suppressor effects in vivo. Surprisingly, a number of studies, including the one reported by Liu et al., show a strong effect of anti-34a. Thus, two of the referees would like to request experiments to reproduce some of the anti-34a experiments reported in the original study, such as Figure 1J. These experiments were central to the conclusions drawn by Liu et al*:

*2A) Systemic miR-34 delivery inhibits prostate cancer metastasis and extends survival (point made in Abstract and results depicted in main Figure 3B, 3D). This is a key experiment that is central to the conclusion, especially given that this work is published in Nature Medicine with an emphasis on translational relevance*.

*2B) Anti-miR34 antagomir increases metastasis relative to a control antagomir (Figure 1J). This is central to the claim that endogenous miR-34a antagonizes prostate cancer metastasis*.

*2C) CD44 is a direct target f miR-34a in prostate cancer (Figure 4D). The authors show that miR-34a reduces luciferase activity in a reporter assay and that mutagenesis of the miRNA complementary sites abolish this repression. This is another key conclusion of the paper and should be replicated*.

We agree that the existing experiments only examine the effect of overexpression of miR-34a. Thus, we have added an additional experiment relating to the identification of CD44 as a target of miR-34a; Figure 4D. Please see Protocol 7 of the updated manuscript for details on this experiment.

*3) Given that the sample sizes* can *be as small as 2 or 3 (for example, protocol 3), I suggest the investigators to pursue exploratory examination of data and possibly non-parametric comparison methods, in addition to the proposed ANOVA methods, at the time of data analysis.*

Thank you for this suggestion. At the time of analysis, we will perform the Shapiro–Wilk test and generate a quantile–quantile (q–q) plot to attempt to assess the normality of the data and also perform Levene’s test to assess homoscedasiticity. If the data appears skewed, we will attempt a transformation in order to proceed with the proposed statistical analysis and possibly perform the appropriate non-parametric test. We have also updated the manuscript to address this point.